# Mendelian segregation and high recombination rates facilitate genetic analyses in *Cryptosporidium parvum*

Abigail Kimball[1], Lisa Funkhouser-Jones[1¤a], Wanyi Huang[1], Rui Xu[1¤b], William H. Witola[2], L. David Sibley[1]*

1 Department of Molecular Microbiology, Washington University School of Medicine, St. Louis, Missouri, United States of America, 2 Department of Pathobiology, University of Illinois Urbana-Champaign College of Veterinary Medicine, Urbana, Illinois, United States of America

¤a Current address: Department of Microbial Pathogenesis & Immunology, Texas A&M School of Medicine, Bryan, Texas, United States of America
¤b Current address: State Key Laboratory for Animal Disease Control and Prevention, South China Agricultural University, Guangzhou, China
* sibley@wustl.edu

**Data Availability Statement:** The authors confirm that all data underlying the findings are fully available without restriction. All relevant data are within the paper and its Supporting Information

## Abstract

Very little is known about the process of meiosis in the apicomplexan parasite *Cryptosporidium* despite the essentiality of sex in its life cycle. Most cell lines only support asexual growth of *Cryptosporidium parvum* (*C. parvum*), but stem cell derived intestinal epithelial cells grown under air-liquid interface (ALI) conditions support the sexual cycle. To examine chromosomal dynamics during meiosis in *C. parvum*, we generated two transgenic lines of parasites that were fluorescently tagged with mCherry or GFP on chromosomes 1 or 5, respectively. Infection of ALI cultures or *Ifngr1*$^{-/-}$ mice with mCherry and GFP parasites resulted in cross-fertilization and the formation of "yellow" oocysts, which contain 4 haploid sporozoites that are the product of meiosis. Recombinant oocysts from the $F_1$ generation were purified and used to infect HCT-8 cultures, and phenotypes of the progeny were observed by microscopy. All possible phenotypes predicted by independent segregation were represented equally (~25%) in the population, indicating that *C. parvum* chromosomes exhibit a Mendelian inheritance pattern. The most common pattern observed from the outgrowth of single oocysts included all possible parental and recombinant phenotypes derived from a single meiotic event, suggesting a high rate of crossover. To estimate the frequency of crossover, additional loci on chromosomes 1 and 5 were tagged and used to monitor intrachromosomal crosses in *Ifngr1*$^{-/-}$ mice. Both chromosomes showed a high frequency of crossover compared to other apicomplexans with map distances (i.e., 1% recombination) of 3–12 kb. Overall, a high recombination rate may explain many unique characteristics observed in *Cryptosporidium* spp. such as high rates of speciation, wide variation in host range, and rapid evolution of host-specific virulence factors.

files, and source data are provided as a
supplementary file.

**Funding:** This work was supported by a grant from
NIH (AI175150) to LDS and the work was partially
supported by the National Science Foundation
Graduate Research Fellowship to AKK. The funders
had no role in study design, data collection and
analysis, decision to publish, or preparation of the
manuscript.

**Competing interests:** The authors have declared
that no competing interests exist.

## Author summary

Although sex is essential for the transmission and maintenance of infection of *Cryptosporidium*, it has been historically challenging to study the process of meiosis in this medically relevant protist. We utilize recent methodological advances such as a specialized *in vitro* culture system, cell sorting, and the generation of transgenic parasites to cross identical strains of parasites in the absence of selection pressure to identify intrinsic chromosome behavior during meiosis. By specifically examining the phenotypes from the first generation of parasites, we reveal that cross-fertilization frequently occurs in parasite populations, chromosomes segregate in a Mendelian manner, and the rate of crossover is high on Chromosomes 1 and 5. Understanding these baseline meiotic mechanisms is essential for planning and interpreting future genetic studies of *Cryptosporidium* seeking to identify genes associated with phenotypes of interest.

## Introduction

Infection with apicomplexan parasite *Cryptosporidium* spp. is the second leading cause of diarrheal disease in children under 5 in resource-poor nations and is a significant threat to immunocompromised adults globally [1–3]. The parasite is transmitted through the fecal-oral route and completes its entire life cycle in the intestine of a single host. After excysting from the infectious oocyst, sporozoites invade the host cells, develop into trophozoites, and then undergo multiple rounds of mitotic division to form meronts [4,5]. After three rounds of merogony, merozoites develop into either male (microgamonts) or female (macrogamonts) gametes [6,7]. Males undergo multiple rounds of replication culminating in the release of 16 extracellular males, which locate and fertilize the females resulting in a zygote [4,8]. The zygote then undergoes meiosis and transitions into an oocyst, which contains four haploid progeny (sporozoites). Oocysts are then released into the environment to infect new hosts; thus, the transmission of *Cryptosporidium* is reliant on the production and excretion of meiotically derived oocysts.

Beyond oocyst transmission, sexual reproduction contributes to *Cryptosporidium* pathogenesis in other ways. First, it aids in the maintenance of productive infections, as oocysts that are retained within the original host restart the infectious cycle [5,8,9]. Moreover, the absence of *de novo* oocyst formation in most *in vitro* culture platforms leads to a rapid decline in parasite growth over time, whereas growth is sustained in systems where sex and oocyst production is supported, suggesting that sex is obligatory in this parasite [5,8,10]. Additionally, meiosis is particularly important to the fitness of facultatively sexual eukaryotic pathogens as it allows for the generation of genetically diverse progeny, which can outcompete and eliminate deleterious offspring that would otherwise be continuously amplified by asexual reproduction [11]. It is likely that currently undefined meiotic mechanisms contribute to the rapid adaptation of *Cryptosporidium* spp. to novel hosts, modulate its virulence, and lead to a large variety of distinct species and subtypes [12–14]. Although studies of *Cryptosporidium* isolates have revealed that introgression frequently occurs between different strains in nature, it is currently unclear how the parasite modulates this exchange during meiosis [14–16].

Meiosis is an ancient invention that has been well-conserved throughout evolutionary time and across the Eukaryotic kingdom. The meiotic process has been studied in a variety of organisms to reveal large variations in molecular mechanisms. These differences can provide insights into the life history of an organism as well as contribute to our overall understanding of the evolution and maintenance of sex. Recently, genomic and molecular studies examining

meiosis in obscure eukaryotes has led to the identification of many novel non-Mendelian mechanisms of chromosomal segregation [17,18]. For instance, *Trypanosoma cruzi* and *Leishmania* spp. parasites have significant genomic plasticity due to frequent chromosomal aneuploidies and random loss of heterozygosity [19–21]. In *Candida albicans*, tetraploid progeny arise from diploid cell mating and is reduced using the non-meiotic "concerted chromosome loss" approach [22], and the binuclear *Giardia* utilizes a specialized process called "diplomixis" during encystation as a mechanism for recombination [23].

Historically, studies of meiosis in protists have been technically challenging due to their microscopic size, lack of culture systems, and underdevelopment of molecular tools. Studies in apicomplexan parasites *Plasmodium* spp. and *Toxoplasma gondii* demonstrate that the parasites are haploid throughout their life cycle, fuse to form a diploid zygote, and undergo meiosis that follows classical Mendelian rules [24–26]. Forward genetic analysis based on classical genetic mapping has enabled the discovery of virulence factors and drug resistance mechanisms in these parasites [24–26]. However, very little is known about how meiosis works in the related apicomplexan parasite *Cryptosporidium*. Adenocarcinoma cell lines (HCT-8 cells) only support the asexual phase of the *Cryptosporidium parvum* (*C. parvum*) life cycle, resulting in short-term parasite growth [8,27]. Reliance on this culture platform has historically limited *C. parvum* research, leaving the processes of fertilization, meiosis, and oocyst formation largely under studied. *C. parvum* is obligately sexual, and the transmission of the parasite to a new host is reliant on the production of meiotically-derived oocysts [8,9,28]. Although studies of natural isolates have revealed that introgression frequently occurs between different strains in nature, it is currently unclear how the parasite modulates this exchange during meiosis [29]. Independent segregation of its 8 linear chromosomes and intrachromosomal recombination likely contribute to the ability of *Cryptosporidium* spp. to rapidly adapt to novel hosts, modulate its virulence, and develop a large variety of distinct species and subtypes [12].

We recently developed a specialized Transwell culture system that uses an air-liquid interface (ALI) to induce differentiation of intestinal stem cells into an epithelial monolayer that permits sexual reproduction of *C. parvum in vitro* [10]. ALI cultures allow *C. parvum* fertilization and oocyst production timing to be precisely predicted, providing a novel opportunity to examine the molecular mechanisms of meiosis in this medically relevant protist. Like *T. gondii* and *Plasmodium* spp., *C. parvum* is thought to be haploid during asexual growth and forms haploid macro- and microgamonts that fuse to form a diploid zygote during its sexual phase [4,5]. In ALI cultures, and in mice, these zygotes undergo meiosis to yield 4 haploid sporozoites within a single oocyst [9,10,30]. Thus, in principle, it should be possible to use classical genetic mapping to study the segregation and recombination of chromosomes during meiosis in *C. parvum*. A previous cross of two genetically distinct *Cryptosporidium* isolates was performed in mice by sequential passage to estimate genetic recombination frequency [31]; however, since each round of infection in mice results in multiple rounds of meiosis, the precise recombination frequency of chromosomes in a single meiotic event has not been defined for *Cryptosporidium*.

*Saccharomyces cerevisiae* is a model yeast commonly used to study meiotic recombination. Analyses of meiosis in *S. cerevisae* usually include examining the phenotypes of progeny from a large population of asci or spores (known as random ascus or spore analysis) and the direct observation of progeny from an individual spore (tetrad dissection) [32]. In this study, we adapted these traditional genetic techniques to study single lineage (F$_1$) *C. parvum* oocysts, produced by crossing isogenically tagged strains (outcrossed oocysts). "Random oocyst analysis" involved observing the frequency of progeny phenotypes at each oocyst hatching site throughout an entire cell culture infected with recombinant oocysts, while "oocyst tetrad dissection" was used to directly interrogate the phenotypes of individual sporozoites from a single oocyst. We found that genetic exchange in *C. parvum* follows a Mendelian inheritance pattern

of independent segregation of chromosomes and that individual chromosomes exhibit high levels of recombination.

## Results

### *C. parvum* utilizes outcrossing *in vitro* and *in vivo*

Previously, we demonstrated that the ALI system supports both self and cross-fertilization of *C. parvum* [10]. Specifically, co-infection of ALI cultures with differentially tagged strains with a fluorescent marker either on chromosome 1 (Δ*uprt*-mCherry) or chromosome 5 (Δ*tk*-GFP) (Fig 1A) resulted in recombinant parasites that expressed both markers, indicative of outcrossing. However, the frequency of outcrossing, which is expected to occur 50% of the time, was lower than expected due to the greater frequency of selfing by the Δ*tk*-GFP strain relative to Δ*uprt*-mCherry [10]. For this study, we utilized the ALI platform to specifically capture the oocysts produced by the first round of fertilization between Δ*uprt*-mCherry and Δ*tk*-GFP parents. Outcrossed "yellow" oocysts could then be used to examine $F_1$ progeny phenotypes at a population and single oocyst resolution. The number of $F_1$ "yellow" outcrossed oocysts produced in ALI is limited due to the high frequency of selfing of the Δ*tk*-GFP parasites. To obtain greater numbers of recombinant oocysts for random oocyst and tetrad analyses, we increased the number of transwells used for experiments and selectively purified double-positive oocysts by fluorescence-activated cell sorting (FACS) before outgrowth infection of HCT-8 cells. While performing FACS, we found that Δ*tk*-GFP oocysts were still dominant in ALI culture (18,042 oocysts) compared to Δ*uprt*-mCherry oocysts (7,208) and recombinant oocysts (1,886 total) at 3 days post infection (dpi) (S1A Fig). Despite filtering harvested ALI samples and using a positive control for oocysts to generate an "all oocysts" gate, the events in the GFP$^+$, mCherry$^+$, and GFP$^+$/mCherry$^+$ oocyst gates likely contain debris as well as oocysts and are not as reliable for quantifying the rates of selfing and outcrossing compared to outgrowth assays. We also found that the recombinant oocysts had two different populations that varied in the ratio of fluorescence signal. The GFP$^{hi}$/mCherry$^{mid}$ (population A) and GFP$^{mid}$/mCherry$^{hi}$ (population C) oocysts occurred about equally in ALI (1,015 vs. 779 oocysts), and we hypothesize that this is due to the macrogamont "mother" contributing more cytoplasmic proteins than the microgamont "father" during oocyst formation [8] (S1A Fig). We also found that collecting oocysts from ALI cultures past 3 dpi resulted in a third population of GFP$^{hi}$/mCherry$^{hi}$ (population B) oocysts, which we predict is the product of mating between $F_1$ yellow macrogamonts with yellow, green, or red males to form $F_2$+ oocysts (S1B Fig). We found a similar pattern when examining oocysts produced *in vivo*: oocysts collected at 3 dpi had undergone a single round of meiosis (S1C Fig) while those collected at 4 dpi or later had started another round (S1D Fig). To focus on chromosome behavior during a single round of meiosis, we confined our analyses to oocysts collected at day 3 both from *in vitro* and *in vivo* infections.

To examine the rate of outcrossing and selfing *in vivo*, we infected *Ifngr1*$^{-/-}$ mice with equal numbers of Δ*uprt*-mCherry and Δ*tk*-GFP oocysts, sacrificed on 3 dpi, harvested the ileums, and enriched for oocysts by bleaching. Oocysts were plated on a human adenocarcinoma cell line (HCT-8) monolayer at a density of 1000–5000 oocysts per coverslip such that progeny descended from a single oocyst would form distinct clusters as asexual expansion progressed. At 15–18 hpi, cultures were fixed, stained, and the mCherry and GFP expression of each cluster was observed by immunofluorescence (IFA) microscopy. Two independent outgrowth experiments found outcrossing to occur 38.2–60.0% of the time during in vivo infections, in contrast to 20.6–27.9% observed in *in vitro* conditions [10]. The average rate of 49.1% outcrossing *in vivo* was consistent with the expected value of 50.0%, suggesting that outcrossing commonly occurs in *C. parvum* populations (Fig 1B).

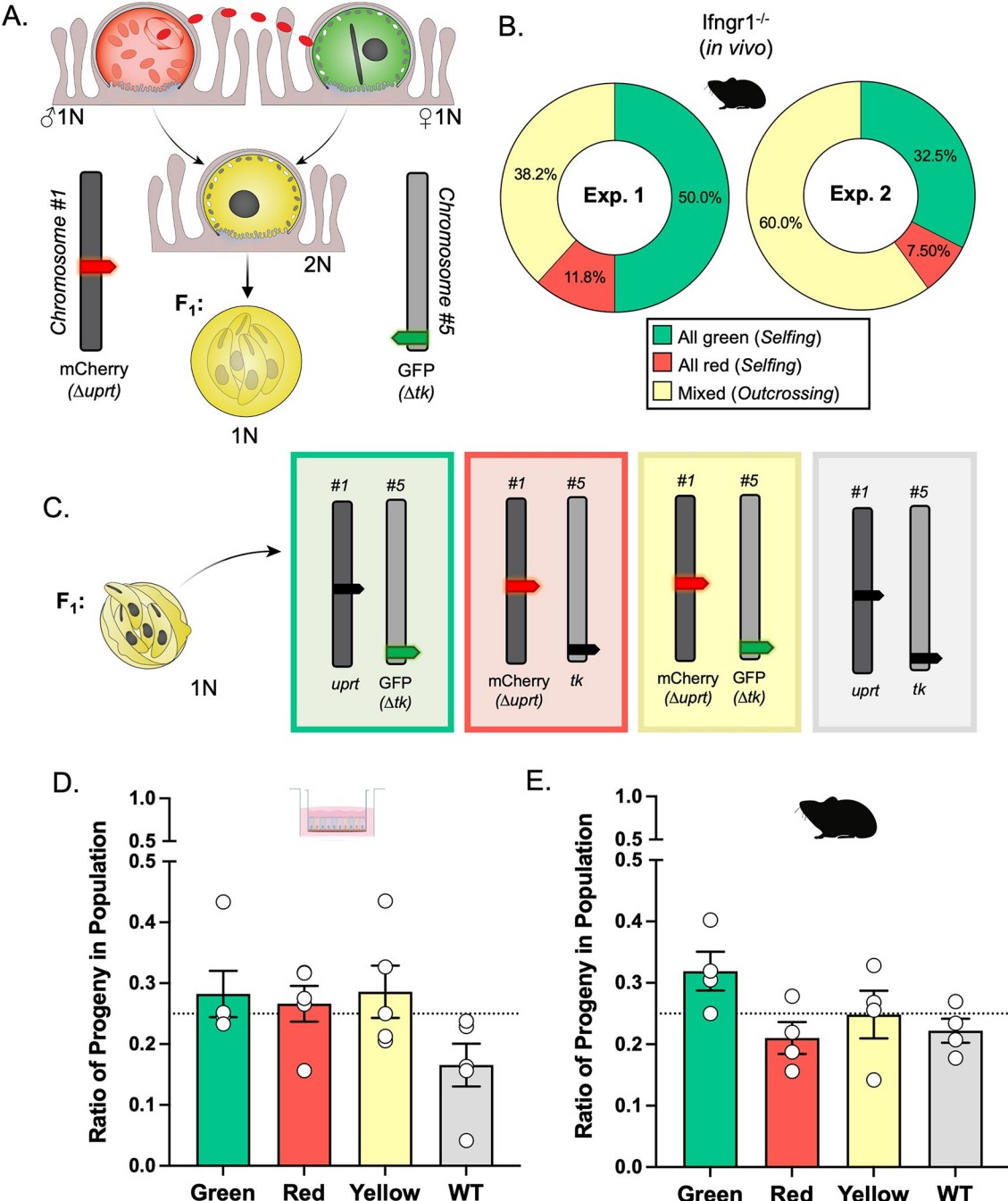

**Fig 1. Detection of outcrossing and the segregation of homologous chromosomes in *C. parvum*.** (A) Parental parasites have chromosome 1 or 5 tagged with mCherry replacing the *uprt* gene or GFP replacing the *tk* gene, respectively. If fusion occurs between gamonts of different phenotypes ("outcrossing"), the resulting progeny will be "yellow" (mCherry+/GFP+). (B) The frequency of $F_1$ outgrowth clusters that were red or green only (selfing) or had more than one color (outcrossing) was determined by immunofluorescence staining and microscopy. Two *Ifngr1*−/− mice infected with Δ*uprt*-mCherry and Δ*tk*-GFP oocysts were sacrificed 3 days post infection (dpi), and $F_1$ "yellow" oocysts were purified from ileal tissue. 1000–5000 $F_1$ "yellow" oocysts were grown on HCT-8 cells, fixed 15–18 hours post infection (hpi) and stained with rabbit anti-GFP, rat anti-mCherry, and VVL-Biotin, followed by secondary antibodies Alexa Fluor 488 goat anti-rabbit IgG, Alexa Fluor 568 goat anti-rat IgG, and Alexa Fluor 647 Streptavidin. Hoechst was used to stain nuclei. This experiment was performed twice and found similar results. (C) Progeny of "yellow" F1 oocysts will have one copy each of chromosomes 1 and 5 inherited from their parents. There are four options for progeny phenotype based on the predicted segregation of chromosomes during meiosis. (D) ALI transwells infected with Δ*uprt*-mCherry and Δ*tk*-GFP oocysts were scraped 3 dpi, bleached, and $F_1$ "yellow" oocysts were purified with FACS. 1000–5000 "yellow" $F_1$ oocysts were grown on HCT-8 cells, fixed, and stained as described above. Data represent the combined mean ± standard error of the mean (SEM) of results of five

independent experiments, with two to three technical replicates per experiment. (E) Two mice were infected, oocysts were collected, and HCT-8 cells were infected, fixed, and stained as described above. All the phenotypes of progeny from "yellow" F1 oocysts in the mixed clusters were quantified. This experiment was performed twice and found similar results. The observed ratios in D and E do not significantly deviate from the expected 25% for each phenotype (p = 0.06 for 1D and p = 0.17 for 1E, Chi-square test) (S1 Dataset).

## Homologous chromosomes segregate independently *in vitro* and *in vivo*

We expect that after oocyst formation, homologous chromosomes segregate from each other so that each sporozoite carries only one copy of each chromosome (either fluorescently tagged or wild type) (Fig 1C). This principle can be tested by examining the population of progeny from recombinant oocysts since each combination of alleles (red, green, yellow, wild type) should be equally represented (25% frequency) in the resulting infection. For *in vitro* experiments, transwells were infected with equal numbers Δ*uprt*-mCherry and Δ*tk*-GFP sporozoites, ALI was harvested 3 dpi, recombinant oocysts were purified by FACS, and 500–5000 yellow oocysts were used to infect HCT-8 cells. At 15–18 hpi, the cultures were fixed and stained, and the progeny phenotypes were quantified by IFA in 5 independent experiments. For *in vivo* experiments, *Ifngr1*[-/-] mice were infected with Δ*uprt*-mCherry and Δ*tk*-GFP oocysts, sacrificed after 3 days, and oocysts were purified from the ileal tissue. All recovered oocysts were used to infect HCT-8 cells, which were fixed and stained after the first round of merogony. Only the progeny in "mixed" clusters were quantified across two experiments. The average distribution of progeny phenotypes (green, red, yellow, and wild type) did not significantly deviate from the anticipated 25% *in vitro* (Fig 1D) or *in vivo* (Fig 1E). Overall, this indicates that homologous chromosomes independently segregate in *C. parvum*.

## *C. parvum* exhibits an unusually high rate of tetratype formation in recombinant oocysts

To further characterize meiosis in *C. parvum*, we tested if homologous chromosomes abided by Mendel's law of independent assortment, which states that genes carried on different chromosomes segregate independently during meiosis. This principle can be tested by examining the progeny from cross-fertilization events at the single oocyst level. Since the *uprt* and *tk* genes are carried on different chromosomes (1 and 5 respectively), each progeny should only receive one copy of either the wild type *uprt* or the Δ*uprt*-mCherry gene and one of the wild type *tk* or Δ*tk*-GFP gene. Depending on how the homologous chromosomes align during metaphase 1, there is an equal probability of an alignment that results in a parental ditype (two "red" and two "green") or non-parental ditype (two "yellow" and two "wild type") progeny pattern. There is also a tetratype inheritance pattern (one of each possible phenotype) that is possible if crossover occurs during interphase I near the *uprt* or *tk* locus. If *C. parvum* abides by the traditional meiotic mechanisms, we would expect all mixed-hatching clusters to fall into one of these categories (Fig 2A).

Analysis of yellow oocysts generated *in vitro* demonstrated that 87.3% of single oocyst outgrowth clusters contained three or four of the possible phenotypes (Fig 2B). The rest of the clusters had various combinations of two colors (PD, NPD) or unexpected combinations (e.g., yellow and red, yellow and green, wild type and red, or wild type and green). This "other" category does not match the parental ditype or non-parental ditype and likely occurs when only some sporozoites successfully egress from the oocyst, establish infection, or go through merogony. We repeated this experiment using yellow oocysts produced in *Ifngr*[-/-] mice and found a similar frequency of TT clusters (89.2%), indicating that this is not a phenomenon specific to *in vitro* culture (Fig 2D). To examine this process in real time, we used recombinant oocysts sorted from ALI cultures to perform live microscopy infection

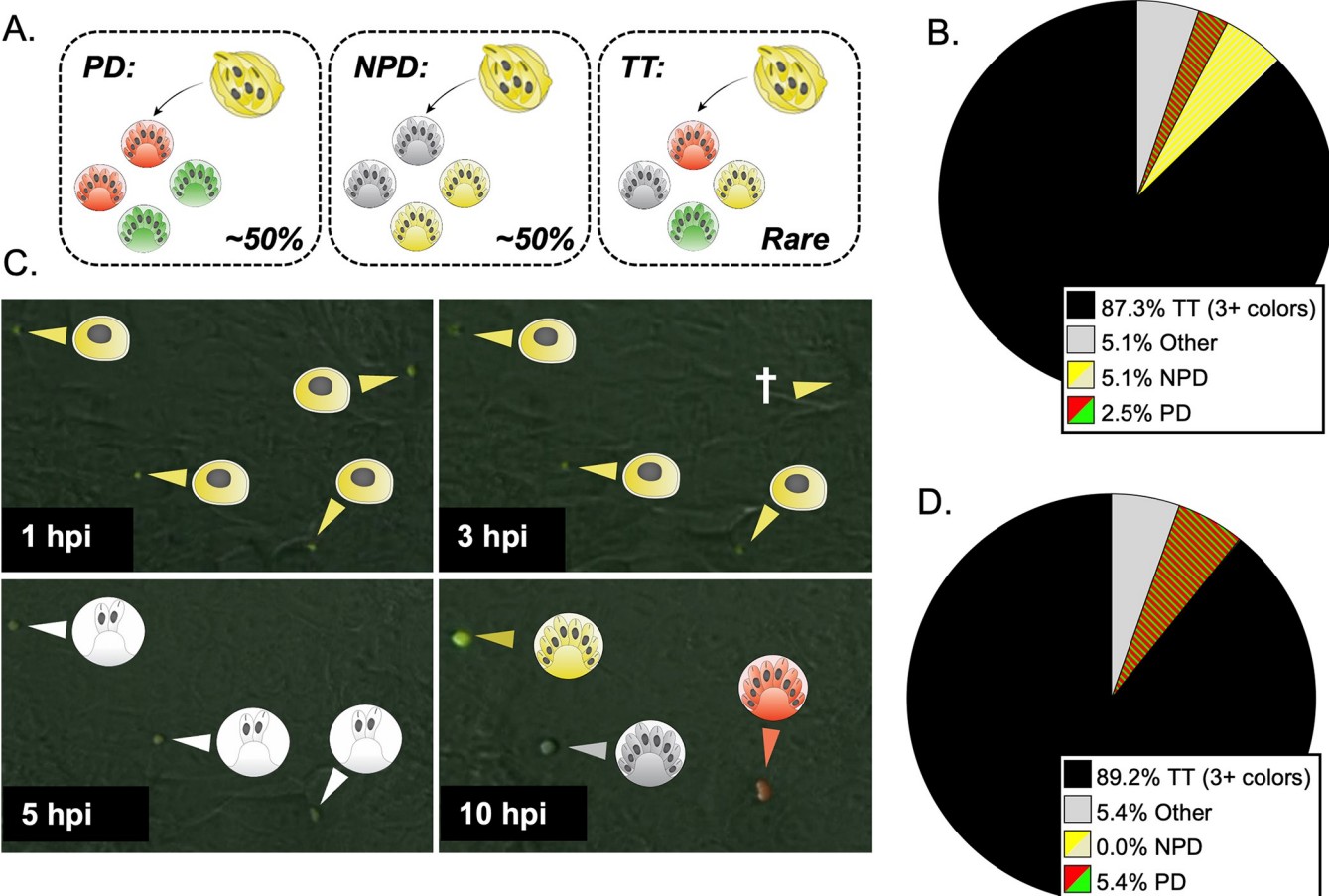

**Fig 2. A tetratype segregation pattern occurs frequently at the single oocyst resolution.** (A) Progeny of "yellow" F1 oocysts are expected to follow one of three segregation patterns: parental ditype (PD, red and green), non-parental ditype (NPD, wild type and yellow), and tetratype (TT, red, green, wild type, and yellow). In the absence of gene linkage and high crossover, you would expect parental ditype and non-parental ditype to occur at a similar rate and tetratype to be rare. (B) The number of mixed clusters that followed the PD, NPD, TT, or other (red and wild type, red and yellow, green and wild type, or green and yellow) segregation pattern was determined by immunofluorescence staining and microscopy. ALI transwells infected with $\Delta uprt$-mCherry and $\Delta tk$-GFP oocysts were scraped and bleached 3 dpi, and F1 "yellow" oocysts were purified with FACS. 1000–5000 F1 "yellow" oocysts were grown on HCT-8 cells, fixed, and stained with the antibodies described in Fig 1. This experiment was performed five times, and 114 mixed clusters were quantified in total. (C) F1 yellow oocysts purified from ALI were used to infect HCT-8 culture, and the expression of GFP and mCherry proteins was captured from oocyst hatching through the conclusion of the first merogony cycle with timelapse microscopy (S1 Movie). Representative images from 1, 3, 5, and 10 hpi are shown. Yellow arrows indicate parasites expressing GFP and mCherry proteins, red arrows indicate the expression of mCherry only, and white arrows indicate parasites that are not expressing GFP or mCherry. The cross symbol indicates a parasite that failed to infect or died before the expression of GFP and/or mCherry could be determined. (D) Two *Ifngr1*$^{-/-}$ mice infected with $\Delta uprt$-mCherry and $\Delta tk$-GFP oocysts were sacrificed 3 dpi, and F1 "yellow" oocysts were purified from ileal tissue. 1000–5000 F1 "yellow" oocysts were grown on HCT-8 cells and fixed 15–18 hours post infection (hpi) and stained as described above. Mixed clusters were examined, and their segregation pattern was quantified (S1 Dataset). This experiment was performed twice, and 34 mixed clusters were quantified total.

experiments in HCT-8 culture. Over three independent experiments, we tracked 292 yellow oocysts from hatching through merogony and found multiple examples of TT segregation (Fig 2C and S1 Movie). Sporozoite and trophozoite stages initially appeared as GFP$^+$/mCherry$^+$ due to the parentally inherited fluorescent cytoplasmic proteins. Early in merogony, these inherited proteins were degraded completely at 5–10 hpi, prior to the parasites expressing fluorescent proteins that reflected their inherited $\Delta uprt$-mCherry and/or $\Delta tk$-GFP genes at 7.5–13.5 hpi (Figs 2C and S2 and S1 Movie). This variation in the timing of protein degradation and expression is correlated with parasite development that is asynchronous between oocyst hatching sites.

### *C. parvum* has a high crossover rate between homologous chromosomes during meiosis

The high frequency of TT inheritance in *C. parvum* was unexpected as similar studies in *T. gondii* found very low rates of recombination between independent markers on separate chromosomes [26,33–35]. In those studies, the pattern of inheritance due to outcrossing closely matched the expected outcome for independent segregation of unlinked markers, which also had low rates of recombination [33]. In contrast, a preponderance of TT genotypes is expected in situations where markers are distal from their centromeres and hence have a high frequency of recombination, as often observed in genetic crosses of *Saccharomyces cerevisiae* [32]. The composition and position of centromeres in *C. parvum* is unknown, and it remained possible that a high rate of recombination on either or both chromosomes was responsible for the TT pattern. To test this possibility, we generated two additional transgenic lines of parasites that had an alternative tagged locus on chromosome 1 or 5 and then performed intrachromosomal crosses and evaluated the rate of crossover. The ABC Transporter gene (*cgd1_700*) has been previously tagged by our lab and shown to have a unique staining pattern in the feeder organelle of all intracellular life stages [36]. It lies on the distal end of chromosome 1, and here we tagged it with CFP using CRISPR/Cas9 (Figs 3A and S3B–S3D). We infected mice as described above with an equal proportion of ABC-3HA-CFP and Δ*uprt*-mCherry parasites and examined the pattern of inheritance in mixed clusters (outgrowth from recombinant "purple" oocysts) at the single oocyst resolution. *Crossover on chromosome 1 would generate a TT pattern at the single oocyst resolution, while absence of crossover should yield PD (Fig 3B)*. We found that the TT pattern was dominant (89.3% of hatching clusters), suggesting that crossover is very common on chromosome 1 (Fig 3C). A similar intrachromosomal crossing experiment was conducted to examine the crossover rate on chromosome 5. The enolase gene (*cgd5_1960*) was tagged with an HA tag (Figs 4A and S4B–S4D) and used in an *in vivo* intrachromosomal cross with the Δ*tk*-GFP transgenic line (Fig 4B). Because the HA tag is not a fluorescent reporter, we fixed and stained with anti-HA antibodies in combination with anti-GFP to examine the outcome. We analyzed the outgrowth from recombinant oocysts (clusters containing both markers) and found that chromosome 5 also has a very high rate of crossover (82.1%) (Fig 4C). To estimate the rate of recombination (cM, 1% recombination) for *C. parvum* we divided the distance between markers (bp) by the percent of clusters that were "TT" or "TT + other" observed over 78–93 outcrossed clusters (Table 1).

## Discussion

Despite evidence for introgression in wild populations, and recent evolution of new variants, little is known about genetic recombination mechanisms during meiosis in *Cryptosporidium*. We developed assays to track meiotic mechanisms by examining $F_1$ progeny from crosses between transgenic parasites *in vitro* and *in vivo*. We found that in our stem cell-derived culture system (ALI) and *Ifngr1*[-/-] mice, transgenic parasites utilize self and cross-fertilization, consistent with the absence of fixed mating types, like other apicomplexans. Examination of progeny in outcrossed oocysts that were allowed to expand mitotically revealed that homologous chromosomes independently segregate *in vitro* and *in vivo*. At the single oocyst level, the segregation pattern of independent markers indicated a high rate of crossing over as tetratype patterns of inheritance predominated. Consistent with this pattern, intrachromosomal crosses between additional lines of transgenic parasites found that crossover is very high on both chromosomes 1 and 5. Our findings indicate that recombination frequently occurs between strains and that high rates of intrachromosomal crossover are common. Such Mendelian patterns of meiotic segregation and high recombination provide a mechanistic explanation for recent

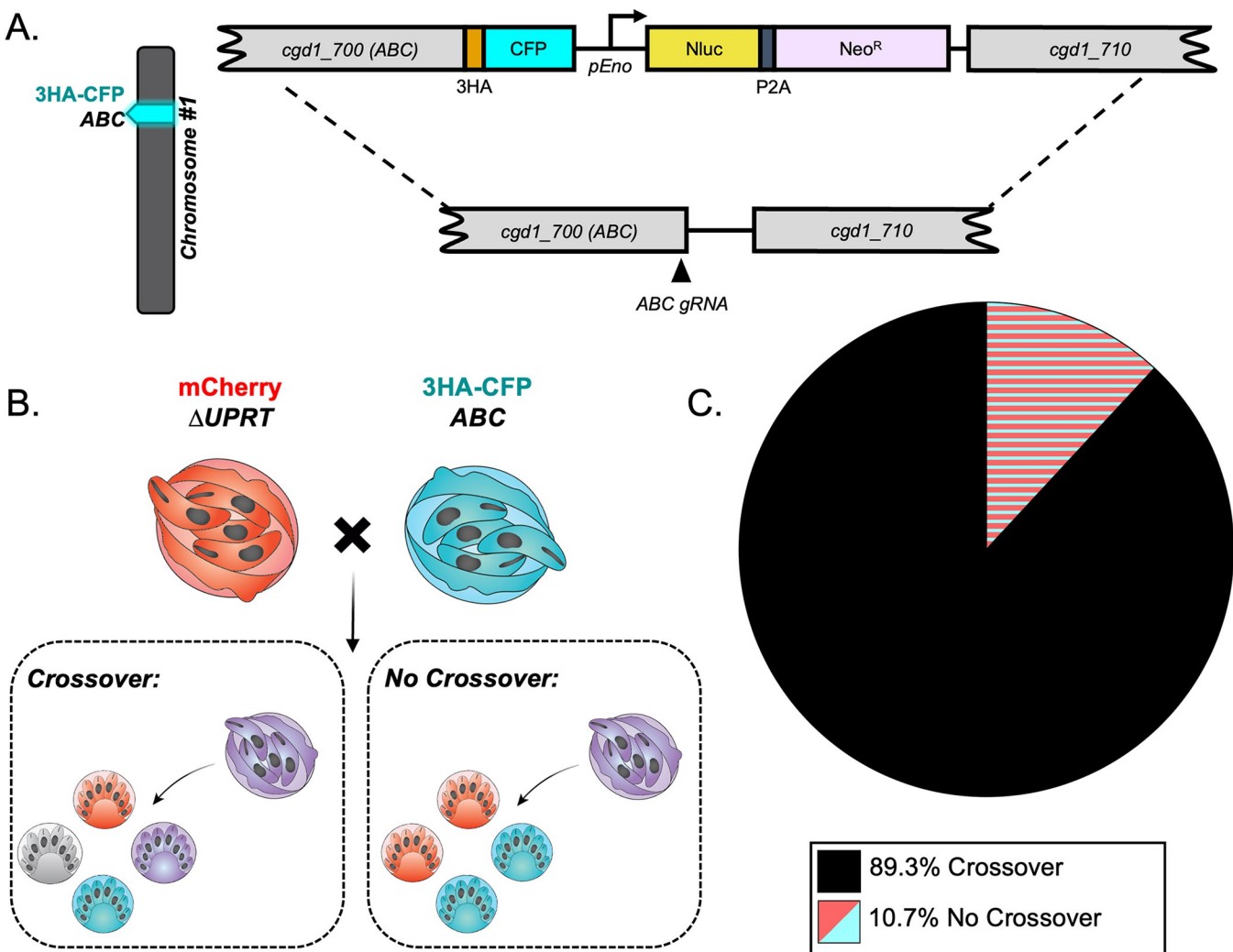

**Fig 3. Crossover is high on chromosome 1.** (A) Diagram of the ABC-3HA-CFP targeting vector. A triple hemagglutinin (3HA) epitope tag, CFP, and an Nluc-P2A-Neo[R] cassette was added to the C-terminus of ABC1 (*cgd1_700*) using CRISPR/Cas9 genome editing. (B) The segregation pattern of progeny from outcrossed oocysts in an intrachromosomal cross reflects if crossover occurred on chromosome 1. A tetratype pattern (mCherry[+], CFP-3HA[+], mCherry[+]/CFP-3HA[+], and wild type) indicates that crossover occurred, whereas a parental ditype pattern (mCherry[+] and CFP-3HA[+]) indicates that no crossover occurred. (C) Two *Ifngr1*[−/−] mice infected with *Δuprt*-mCherry and ABC-3HA-CFP oocysts were sacrificed 3 dpi, and all F_1 oocysts were purified from ileal tissue. 1000–5000 F_1 oocysts were grown on HCT-8 cells and fixed 15–18 hours post infection (hpi) and stained. The antibodies used were rabbit anti-GFP (also recognizes CFP), rat anti-mCherry, and VVL-Biotin, followed by a secondary stain of Alexa Fluor 488 goat anti-rabbit IgG, Alexa Fluor 568 goat anti-rat IgG, and Alexa Fluor 647 Streptavidin. Lastly, Hoechst was used to stain nuclei. Mixed clusters were examined, and their segregation patterns were quantified. This experiment was performed twice, and 93 mixed clusters were quantified in total (S1 Dataset).

recombination and development of new genotypes in the wild including those that are anthro-ponotic subtypes [14].

Other apicomplexan parasites require multiple hosts to complete the asexual and sexual portions of their life cycles, requiring multiple animal models to perform forward genetic crosses [24,26], and precluding genetic crosses *in vitro*. However, *C. parvum* is monoxenous and can be cultured entirely in a specialized ALI culture system *in vitro*, thus allowing genetic crosses to be performed in a few days in the laboratory [10]. Our laboratory previously showed that the ALI system could be used for mating transgenic parasites, but outcrossing only occurred ~25% of the time (vs. the expected 50%), and there was an overrepresentation of

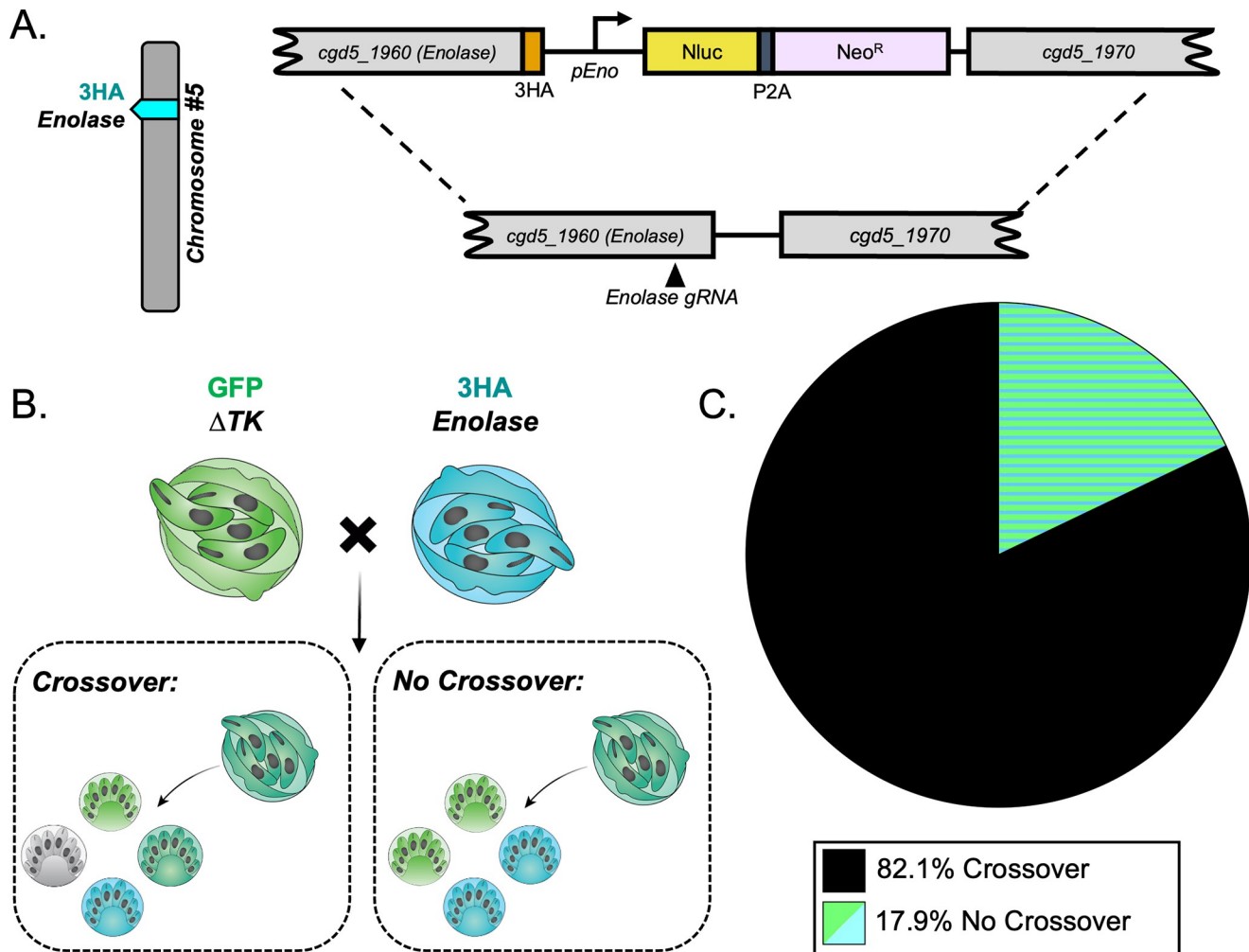

**Fig 4. Crossover is high on chromosome 5.** (A) Diagram of the Enolase-3HA targeting vector. A triple hemagglutinin (3HA) epitope tag and an Nluc-P2A-Neo$^R$ cassette was added to the C-terminus of enolase (*cgd5_1960*) using CRISPR/Cas9 genome editing. (B) The segregation pattern of progeny from outcrossed oocysts in an intrachromosomal cross reflects if crossover occurred on chromosome 5. A tetratype pattern (GFP$^+$, 3HA$^+$, GFP$^+$/3HA$^+$, and wild type) indicates that crossover occurred, whereas a parental ditype pattern (GFP$^+$ and 3HA$^+$) indicates that no crossover occurred. (C) Two *Ifngr1*$^{-/-}$ mice infected with Δ*tk*-GFP and Enolase-3HA oocysts were sacrificed 3 dpi, and all F$_1$ oocysts were purified from ileal tissue. 1000–5000 F$_1$ oocysts were grown on HCT-8 cells and fixed 15–18 hours post infection (hpi) and stained with rabbit anti-GFP, rat anti-HA, and VVL-Biotin, followed by a secondary stain of Alexa Fluor 488 goat anti-rabbit IgG, Alexa Fluor 568 goat anti-rat IgG, and Alexa Fluor 647 Streptavidin. Hoechst was used to stain nuclei. Mixed clusters were examined, and their segregation patterns were quantified. This experiment was performed twice, and 78 mixed clusters were quantified total (S1 Dataset).

green clusters over red. These fluorescently tagged parasites lack either the *tk* gene or one of two copies of *uprt*, yet their growth *in vitro* and *in vivo* are relatively normal [10]. The non-essential nature of these two genes is likely due to alternative pathways: DHFR-TS can provide *de novo* synthesis of pyrimidines in the absence of TK and the second copy of *uprt* is likely sufficient. Nonetheless, the slightly lower success of selfing of Δ*uprt*-mCherry suggests it may have a defect in growth and/or development. This lower selfing rate in Δ*uprt*-mCherry was also observed *in vivo*, although the proportion of outcrossed oocysts was somewhat higher than observed *in vitro*. Selfing and outcrossing is a common pattern seen in other apicomplexans such as *Plasmodium* spp. and *Toxoplasma gondii* [37,38]. Maintaining a 50/50 rate of

**Table 1. Estimated recombination frequencies for *Cryptosporidium parvum* chromosomes 1 and 5.**

| Chromosome | Gene ID | Tag | Position | Distance between markers | Recombination Frequency (1% in bp*) | % of TT | % of TT + other |
|---|---|---|---|---|---|---|---|
| 1 | cgd1_1900 | Δuprt-mCherry | 436,349–437,147 bp | 264,369 bp | 2962–3841 bp | 68.82% | 89.25% |
| 1 | cgd1_700 | ABC-3HA-CFP | 167,685–171,980 bp | | | | |
| 5 | cgd5_4440 | Δtk-GFP | 1,018,878–1,019,486 bp | 603,027 bp | 7349–12,060 bp | 50% | 82.05% |
| 5 | cgd5_1960 | Enolase-3HA | 414,502–41,5851 bp | | | | |

*First number: % TT in X clusters, second number: % TT + "Other" in 78–93 clusters from two independent experiments.

selfing and outcrossing is likely highly advantageous in parasites, where constant genetic innovation, but also preservation, must be delicately balanced for survival.

There are many examples of non-Mendelian inheritance patterns in fungal and protist species (chromosomal aneuploidies with reduction by random loss of heterozygosity or concerted chromosome loss, diplomixis, etc.) [23]. These various systems can each contribute to genetic variability and hence are adaptive. However, understanding the parameters of meiosis is key to using forward genetics to map traits and genes. To examine the behavior of chromosomes during meiosis in *Cryptosporidium*, we purified recombinant oocysts from ALI and immunocompromised mice. Analysis of 162 outcrossed clusters revealed that alleles on homologous chromosomes segregate independently and results in predicted ratios of parental and recombinant progeny *in vitro* and *in vivo*. Confirmation that *C. parvum* independently segregates alleles is essential for performing more advanced genetic studies like gene mapping and linkage analyses.

Studies examining meiosis in model organisms typically rely on cytological examination of the process in zygotes. However, isolation of the diploid stage of *C. parvum* is currently infeasible as zygotes are an extremely rare and transient population, there are no genes uniquely associated with this life stage, and apicomplexan parasites do not condense their very small chromosomes [39]. For these reasons, it was necessary to utilize an indirect approach where we examined the phenotypes of progeny from outcrossed oocysts to deduce the meiotic process. Based on previous experiments in *T. gondii* [26,33–35] we initially expected that the parental ditype and non-parental ditype pattern would be equally common and dominate culture, and the tetratype pattern would occur occasionally. However, we found the opposite to be true as the tetratype segregation pattern was preponderant, indicating to us that the crossover rate on chromosome 1 and/or 5 is relatively high.

The rate of crossover in *C. parvum* was quantified previously [31], but at that time, recombinant lines could not be created, and the generation number of progeny could not be verified ($F_1$ vs. $F_2$ and beyond.). For our study, we generated two additional lines of parasites that had alternative loci on chromosome 1 and 5 tagged and performed an intrachromosomal cross. We also only examined the progeny of "yellow" F1 oocysts" to observe the frequency of crossovers between markers in a single meiosis (analogous to a random spore analysis in yeast). We found a high crossover rate for both chromosomes 1 and 5, although without sequencing the progeny it is unclear if crossovers occur randomly between the markers or if they are clustered. We found an overall recombination rate of 3–3.8 kb (Chromosome 1) and 7.4–12.1 kb (Chromosome 5) per centimorgan that is slightly lower than a previous estimate of 10–56 kb/cM for *C. parvum* [31]. Our estimated recombination rates apply between discrete markers on two chromosomes, while this previous rate was a genome wide estimate and rates could vary

considerably between chromosomes. Our calculated recombination rate could be an underestimate as our genetic markers are far enough away from one another that double or an even number of crossover events would not be detected by our assay (Table 1). The recombination rate for *C. parvum* is roughly similar to that of *P. falciparum* 9.6–17 kb/cM [40] but higher than the ~104 kb/cM in *T. gondii* [34]. In comparison, the recombination rate in *Cryptococcus neoformans* is estimated to be 13.2 kb/cM [41], 260 kb/cM in *Arabidopsis thaliana* [42], and 137 kb/cM in *Drosophila pseudoobscura* [43]. In model organisms, the recombination rate can be influenced by a variety of factors such as the size of the chromosome, the centromere position, crossover assurance, etc. [44,45]. It is currently unknown what type of centromeric structure *C. parvum* utilizes, where centromeres are located on each chromosome, and what effects proximity has on recombination rates. Furthermore, crossover assurance (when at least one crossover must occur per chromosome during chiasma formation between homologous chromosomes during meiosis) does not occur in every organism. For instance, many *T. gondii* chromosomes do not undergo crossover in individual progeny from outcrosses [35]. Our current data do not allow us to calculate crossover events over all chromosomes, although future studies with additional marker or genome-wide analysis could address this issue.

Our findings are consistent with the many observations of introgression between *Cryptosporidium* spp. variants [14,16] as sexual reproduction likely frequently occurs between parasites cohabitating in a single host. Previous comparative genomics studies from wild *Cryptosporidium* spp. populations have observed that frequency of recombination events varies significantly between chromosomes with disproportionately lower (chromosomes 3, 5, and 7), moderate (chromosomes 1, 2, 4, and 8), and high (chromosome 6) number of events [14]. However, these identified recombination events come from comparing sequencing data from a variety of subtypes and species, whereas our study is the first to perform a cross between identical strains in the absence of selection pressure or bias to capture the intrinsic chromosome dynamics.

There are many inherent traits of *Cryptosporidium* that make it an ideal candidate for genetic analysis such having a single host life cycle, the ability to isolate the oocyst life stage via bleaching, the relative synchronicity of the life cycle, and even the sexual cycle block in HCT-8 cells. Additionally, a high recombination rate will make it easier to identify specific regions of the genome associated with phenotypes of interest. Future crossing studies in *Cryptosporidium* can mate strains of varying virulence with known SNPs to identify genes of interest that can be characterized and utilized as targets for therapeutics. The development of many new tools for the study of *C. parvum* in the last decade has allowed more complex study of this organism [5]. By combining CRISPR/Cas9 with ALI, we were able to adapt the classical yeast genetic assays random ascus/spore analysis and tetrad dissection for the study of recombination in *C. parvum*. Our study sought to establish basic mechanisms of the meiotic process in *C. parvum* by examining the inheritance of specific loci at the phenotypic level without sequencing. Now that we have established that outcrossing is common, alleles segregate in a Mendelian matter, and that recombination and linkage experiments are feasible, a new era of complex genetic screens in *Cryptosporidium* can begin.

## Materials and methods

### Ethics statement

Animal studies on mice were approved by the Institutional Animal Studies Committee (School of Medicine, Washington University in St. Louis). *Ifngr1*[-/-] mice (003288; Jackson Laboratories), and Nod scid gamma mice (referred to as NSG) (005557; Jackson Laboratories) were bred in-house in a specific-pathogen-free animal facility on a 12:12 light-dark cycle. Male and

female mice between 8 and 12 weeks of age were used for sex-matching in crossing experiments. Mice were co-housed with siblings of the same sex throughout the experiments.

## Cell culture

HCT-8 cells (Human ileocecal adenocarcinoma, ATCC CCL-244) derived from a human ileocecal carcinoma from a 67-year-old man were cultured in RPMI 1640 ATCC Modification medium supplemented with 10% fetal bovine serum at 37 °C in a 5% CO2 incubator. After trypsinization, 450,000 cells were plated on glass coverslips in 24-well plates and used for outgrowth assays or IFA studies of transgenic parasite lines. Cells were confirmed to be mycoplasma-free with the e-Myco plus Mycoplasma PCR detection kit.

## Genetic crosses in the air liquid interface (ALI) culture system

**Generating air liquid interface (ALI) culture system.** ALI was generated according to previously described protocols [10,46]. Briefly, transwells were coated with a layer of diluted Matrigel and incubated for 15–20 min at 37°C. Excess Matrigel was aspirated, and irradiated 3T3 (i3T3) cells were seeded at a density of $8 \times 10^4$ cells/transwell at 37°C for 24 hr. Next, mouse ileal spheroids were harvested from culture, processed to a single cell suspension, and $5 \times 10^4$ cells/transwell were added to the i3T3 cell monolayer. Every other day, the top and bottom transwell compartments were fed 50% conditioned media (CM) for 7 days total [10,46]. After 7 days of growth, the top media was removed for an additional three days before infection while the bottom media continued to be changed every other day throughout the experiment.

**Infecting ALI with *Cryptosporidium* parental oocysts.** The infection of ALI transwells has been previously described by our lab [10,46,47]. Before infection of ALI transwells, transgenic strains of parasites were treated with a 40% bleach solution for 10 min on ice, washed 3 times in DPBS w/ 1% BSA, and resuspended in DPBS w/ 1% BSA to be at $1 \times 10^8$ oocysts/mL for excystation. Excystation buffer (1.5% sodium taurocholate) was added at a 1:1 concentration, and oocysts were incubated at 37°C for one hr. Sporozoites were then spun down and resuspended in 50% CM, and $2 \times 10^5$ sporozoites of each parental transgenic strain were added per transwell ($4 \times 10^5$ total sporozoites/transwell). After 4–6 hr, the media and uninfected parasites were removed, and the transwells were washed twice with DPBS.

**Harvesting $F_1$ oocysts from ALI.** After three days of infection, the transwells were moved into empty wells of a 24-well plate, 100 uL of cold DPBS was added to each well, and a blunt pipette tip was used to scrape the monolayer off the transwell. 100 uL from each transwell was then transferred to a polystyrene 15 mL conical tube and treated with 40% bleach solution for 10 min on ice. Oocysts were then spun down at 1,250 x g for 3 min, washed twice with DPBS, and used for sorting experiments.

**Fluorescent activated cell sorting (FACS) of yellow oocysts from ALI.** The SONY SH800S Cell Sorter software and control samples were used to construct a Boolean gating strategy to identify oocysts (FSC-A$^{lo}$ and BSC-A$^{lo}$) and "yellow oocysts" (GFP$^+$/mCherry$^+$). Oocysts harvested from ALI were resuspended in 100 uL of cold DPBS and inserted into the sorting chamber of a SONY SH800 Cell Sorter, and 1000–5000 yellow oocysts were sorted directly into a 24-well plate containing coverslips with a confluent HCT-8 cell monolayer (2–3 coverslips total per experiment) or into 100 uL of cold DPBS for storage.

**Timelapse microscopy.** Timelapse microscopy of *C. parvum* infection has been previously described by our lab [48]. HCT-8 cells were cultured in a poly-lysine coated glass-bottomed 24-well plate for 24 hr before the experiment. Media was replaced with HCT-8 media (with 10% FBS and no phenol red) containing ~1000 sorted yellow oocysts. ~100 oocysts were

tracked every 20 min for 30 hr at 40X magnification using a Zeiss Observer Z1 471 inverted microscope with a Colibri LED illumination for multi-color epifluorescence, and the ORCA-ER digital camera was used for image acquisition. All images were processed and analyzed using ZEN 2.5 software.

**Immunofluorescence assays & microscopy.** Staining protocols have been previously described [49]. Briefly, infected HCT-8 cultures were fixed with 4% formaldehyde 16–18 hr post-infection (hpi), permeabilized with 0.1% Triton X-100, blocked with 1% bovine serum albumin (BSA), stained with a primary antibody mixture for 1 hr followed by secondary antibodies for 1 hr and Hoechst for 20 min. Primary antibodies were diluted for staining: 1E12 (mouse, 1:500) [49], VVL-Biotin (*Vicia Villosa* Lectin, Biotinylated, 1:1000), anti-GFP (rabbit, 1:1000), anti-mCherry (rat, 1:1000), and anti-HA (rat, 1:500). All Alexa Fluor secondary antibodies were diluted at 1:1000 and Hoechst was diluted 1:2000. Coverslips were mounted onto microscope slides with Prolong Glass Antifade Mountant. Coverslips were visualized using the Zeiss Axioskop Mot Plus fluorescence microscope with 63X and 100X oil objectives. Images were taken with the AxioCam MRm monochrome digital camera and Axiovision software and further analyzed using ImageJ.

**Crossing experiments in immunodeficient mice.** Parental oocyst strains were prepared for mouse infections ($1 \times 10^6$–$2 \times 10^6$ total oocysts/mouse) using bleaching and washing methods described previously [50], and *Infgr1*$^{-/-}$ mice were orally gavaged. After 3 days post-infection, mice were euthanized by $CO_2$ asphyxiation and the ileum was harvested into a petri dish containing 10 mL of cold DPBS. The submerged ileum was cut open longitudinally and the tissue and liquid were transferred into a 15 mL polystyrene conical tube. The samples were placed on a nutating rocker for 10 min, the contents were poured into a 50 mL polystyrene conical tube through a 40 μm filter, and an additional 30 mL of cold DPBS was added to flush the filter before disposing of the tissue. The samples were spun down at 1,250 x g for 3 min, and the oocysts were bleached and washed as described above. 1000–5000 oocysts per coverslip were resuspended in HCT-8 media and used for infection (2 coverslips total per mouse).

**CRISPR/Cas9 plasmid construction.** The pABC-3HA-Nluc-P2A-neo homology repair template and the pACT:Cas9, U6:sgABC plasmids were previously generated in our lab [36]. For this study, an additional CFP fluorescent protein was inserted into the repair template via Gibson assembly cloning with the linearized backbone and a gBlock gene fragment (Integrated DNA Technologies). The Enolase CRISPR/Cas9 plasmid was created via Gibson assembly of the linearized backbone amplified from the pACT:Cas9, U6:sgABC plasmid and an oligonucleotide with the sgRNA specific for the Enolase gene. The Enolase tagging plasmid was generated using the pABC-3HA-Nluc-P2A-neo homology repair template and replacing the C-terminal and 3'UTR sequences with equivalent regions from the Enolase gene amplified from *C. parvum* genomic DNA. SnapGene (v7.1.1) and NEBuilder Assembly Tool (v2.9.0) were used to construct plasmids and design primers and the Eukaryotic Pathogen CRISPR guide RNA/ DNA Design tool (http://grna.ctegd.uga.edu) was used to identify the Enolase sgRNA sequence. Primers were ordered from IDT and all sequences can be found in S2 Table.

**Amplifying transgenic parasites in mice.** The transfection and amplification of parasites in immunocompromised mice have been previously described by our lab [10,36,50,51]. Briefly, $1 \times 10^8$ freshly excysted sporozoites were electroporated with 100 μg of the repair plasmid, and 66 μg of the CRISPR/Cas9 plasmid and resuspended in 300 uL of cold DPBS. *Ifngr1*$^{-/-}$ mice were orally gavaged first with 8% (wt/vol) sodium bicarbonate to neutralize the stomach acid, followed by the electroporated sporozoites. The drinking water was replaced with paromomycin $H_2O$ (16 g/L) 24 hours later to select for transgenic parasites. Fecal pellets were collected for luciferase assays, qPCR, and PCR on 3, 9, and 15 dpi to confirm parasite growth and correct gene insertion. Oocysts were enriched from the fecal material with bleach treatment and

20,000 oocysts/mouse were used for a second round of amplification in NSG mice. Fecal samples were collected every 3 days after 3 dpi and used for the assays described above and for downstream purification for crossing experiments.

**Luciferase assays for mouse experiments.** Fecal samples collected from *Ifngr1*[-/-] and NSG mice were analyzed with the Promega Nano-Glo luciferase assay kit as previously described [50]. Tubes for fecal collection were weighed before and after collection to determine fecal sample weight. Glass beads were added to the sample along with a fecal lysis buffer and vortexed thoroughly. Samples were then centrifuged, supernatant was extracted, and mixed with the Promega Nano-Glo luciferase assay kit buffer and substrate. Luciferase values were then read on the Cytation 3 cell imaging multi-mode reader, and the number of relative luminescence units per milligram of feces was calculated by dividing the average values of two technical replicates by the weight of the fecal sample.

**Quantifying oocyst shedding in mouse experiments.** The QIAamp PowerFecal DNA kit was used to extract DNA from fecal pellets collected from *Ifngr1*[-/-] and NSG mice using previously described methods [10,50,51]. The number of oocysts at each timepoint was determined by qPCR analysis using the *C. parvum* glyceraldehyde-3-phosphate dehydrogenase (GAPDH) primers (S1 Table) and a standard curve based on a dilution series using previously described methods [10]. The amount of gDNA was quantified using the QuantStudio 3 real-time PCR system, and the number of oocysts per milligram of feces was calculated by dividing the average values of two technical replicates by 4 (an oocyst contains 4 parasites) and further dividing by the weight of the fecal sample.

**Confirmation of insertions in transgenic parasites by PCR.** Extracted gDNA from each murine fecal sample on 27 or 30 dpi and a wildtype control was used as the template to confirm the proper integration of the homology repair template by PCR. 1 μL of extracted gDNA, Q5 Hot Start high-fidelity 2x master mix, and primers at a final concentration of 500 nM (S2 Table) were used for PCR amplification on a Veriti 96-well thermal cycler. For the amplification of the 3'UTR in the ABC-3HA-CFP samples and both the 5' and 3'UTR amplification in Enolase-3HA the following cycling condition was used: 98°C for 30 s, followed by 35 cycles of 98°C for 15 s, 65°C for 30 s, and 72°C for 1 min, with a final extension of 72°C for 2 min. For the amplification of the 5'UTR in ABC-3HA-CFP samples the extension time was adjusted to 1.5 min. All PCR products were resolved on a 1% agarose gel containing GelRed (1:10,000) and imaged using the ChemiDoc MP imaging system.

**Oocyst purification from fecal material and storage.** The *Cryptosporidium parvum* isolate AUCP -1 used for transfections was purified from fecal material collected from male Holstein calves at the University of Illinois at Urbana-Champaign [52]. All calf procedures were approved by the Institutional Animal Care and Use Committee (IACUC). Transgenic oocysts were purified in-house from fecal material collected from NSG mice using a saturated sodium chloride solution for oocyst floatation [53]. After purification, oocysts were stored at 4 °C in phosphate-buffered saline (PBS) for up to three months after fecal collection.

## Statistical analysis

All statistical analyses were performed in GraphPad Prism 10. A Chi-squared test was used to compare the expected and observed ratios for progeny phenotypes. Statistical parameters for each experiment including statistical test used, technical replicates (n), and independent biological replicates (N) are reported in the figure legends and associated method details.

## Supporting information

**S1 Fig. Yellow oocyst populations vary in GFP and mCherry expression *in vitro* and *in vivo* on different days of infection.** (A) ALI was infected with $\Delta uprt$-mCherry and $\Delta tk$-GFP oocysts, after 3 days post infection (dpi) transwells were scraped, bleached, and the expression of F1 oocysts were examined by flow cytometry. The $\Delta tk$-GFP oocysts produced by self-fertilization are more frequent than $\Delta uprt$-mCherry oocysts or outcrossed "yellow" oocysts. Yellow oocysts (mCherry$^+$/GFP$^+$) have two different populations that vary slightly in expression. GFP$^{hi}$/mCherry$^{mid}$ and GFP$^{mid}$/mCherry$^{hi}$ yellow oocysts occur at similar rates and likely differ based on the genotype of macrogamont"mother" of the oocyst. (B) A third population of mCherry$^+$/GFP$^+$ oocysts (GFP$^{hi}$/mCherry$^{hi}$) can be found in ALI cultures harvested at 4 days post infection using methods described above. These oocysts are likely the result of F$_1$ yellow macrogamonts mating with green and red microgamonts and are indicative that a second round of meiosis has begun. (C) The GFP$^{hi}$/mCherry$^{mid}$ and GFP$^{mid}$/mCherry$^{hi}$ yellow oocyst populations can also be found in samples collected from Ifngr1$^{-/-}$ mice infected with $\Delta uprt$-mCherry and $\Delta tk$-mGFP harvested after 3 days of infection. (D) Similarly, a third population of GFP$^{hi}$/mCherry$^{hi}$ yellow oocysts can be found in samples collected from mice at 4 days post infection using methods described above. To capture only F$_1$ oocysts all experiments utilized only oocysts harvested at 3 dpi.
(TIF)

**S2 Fig. Parentally inherited cytosolic fluorescent proteins are degraded before germline encoded fluorescent proteins are expressed in progeny from F$_1$ 'yellow' oocysts.** Yellow bars represent the time (hours post infection) where parentally inherited cytosolic fluorescent proteins were expressed in F$_1$ 'yellow' progeny and black dots indicate when these parentally inherited proteins were degraded. The colored dot represents the time when the germline encoded fluorescent protein began to be expressed and were continuously expressed throughout the first phase of merogony. The progeny from an individual F$_1$ "yellow" oocyst were analyzed across 5 videos in two independent experiments. Each row represents progeny from an individual F$_1$ "yellow" oocyst (n = 5) across two independent experiments (S1 Dataset).
(TIF)

**S3 Fig. The ABC gene was successfully tagged in *Cryptosporidium parvum*.** (A) A diagram of the ABC-3HA-CFP targeting vector and the expected sizes for the 5' Ins (1419 bp) and 3' Ins (501 bp) amplicons used for the confirmation of proper insertion by PCR. (B) PCR analysis of ABC-3HA-CFP oocysts amplified in four NSG mice. WT, wild type. M1-4, DNA purified from fecal samples collected from each mouse used for amplification. The 5′ Ins and 3' Ins products are specific for the 5′ and 3' insertion sites of the integrated construct. Primers are identified in S2 Table. (C) ABC-HA-CFP oocysts purified from mouse feces were used for infection of HCT-8 cells, fixed at 18 hpi and stained with rabbit anti-GFP (also recognizes CFP), rat anti-HA, and VVL-Biotin all at 1:500. Followed by a secondary stain of Alexa Fluor 488 goat anti-rabbit IgG, Alexa Fluor 568 goat anti-rat IgG, and Alexa Fluor 647 Streptavidin all at a 1:1000 dilution. Hoechst was used for nuclear staining at 1:2000. (C) Fecal pellets collected from NSG mice from 3 to 30 days post infection were used for nanoluciferase assays and expressed as relative luminescence units (RLU) per milligram of feces. Each point represents a single sample from an individual mouse (n = 4). (D) DNA purified from fecal pellets collected during amplification were used for a qPCR analysis and expressed as gDNA equivalents per milligram of feces. Each point represents a single sample from an individual mouse (n = 4).
(TIF)

**S4 Fig. The Enolase gene was successfully tagged in *Cryptosporidium parvum*.** (A) A diagram of the Enolase-3HA targeting vector and the expected sizes for the 5' Ins (749 bp) and 3' Ins (517 bp) amplicons used for the confirmation of proper insertion by PCR. (B) PCR analysis of Enolase-3HA oocysts amplified in three NSG mice. WT, wild type. M1-3, DNA purified from fecal samples collected from each mouse used for amplification. The 5′ Ins and 3' Ins products are specific for the 5′ and 3' insertion sites of the integrated construct. Primers are identified in S2 Table. (C) Enolase-HA oocysts purified from mouse feces were used for infection of HCT-8 cells, fixed at 18 hpi and stained with rat anti-HA at 1:500 and VVL-Biotin at 1:500. Followed by a secondary stain of Alexa Fluor 488 goat anti-rat and Alexa Fluor 647 Streptavidin at a 1:1000 dilution. Hoechst was used to stain nuclei at 1:2000. (C) Fecal pellets collected from NSG mice from 3 to 27 days post infection were used for nanoluciferase assays and expressed as relative luminescence units (RLU) per milligram of feces. Each point represents a single sample from an individual mouse (n = 3). (D) DNA purified from fecal pellets collected during amplification were used for a qPCR analysis and expressed as gDNA equivalents per milligram of feces. Each point represents a single sample from an individual mouse (n = 3).
(TIF)

**S1 Movie. Progeny from outcrossed oocysts infect HCT-8 cells and display a tetratype segregation pattern.** Freshly excysted F1 progeny from outcrossed oocysts infect HCT-8 cells and move through first round of merogony as captured by video time-lapse microscopy. Cells were grown on HCT-8 cells at 37°C. Images were acquired every 20 min for 30 hours.
(MP4)

**S1 Table. Key reagents and resources used in the experiments.**
(DOCX)

**S2 Table. All primers used for the construction of the ABC-HA-CFP and Enolase-HA transgenic parasites.**
(DOCX)

**S1 Dataset. Source data files for Figs 1–4 and S2 and Table 1.**
(XLSX)

## Acknowledgments

We thank members of the Sibley laboratory for their helpful suggestions.

## Author Contributions

**Conceptualization:** Abigail Kimball, Lisa Funkhouser-Jones, L. David Sibley.

**Data curation:** Abigail Kimball, Lisa Funkhouser-Jones.

**Formal analysis:** Abigail Kimball.

**Funding acquisition:** Abigail Kimball, L. David Sibley.

**Investigation:** Abigail Kimball, Lisa Funkhouser-Jones, Rui Xu.

**Methodology:** Abigail Kimball, Lisa Funkhouser-Jones, Wanyi Huang, Rui Xu, William H. Witola, L. David Sibley.

**Project administration:** L. David Sibley.

**Resources:** Abigail Kimball, Lisa Funkhouser-Jones, Wanyi Huang, Rui Xu, William H. Witola, L. David Sibley.

**Supervision:** L. David Sibley.

**Validation:** Abigail Kimball, Lisa Funkhouser-Jones, L. David Sibley.

**Visualization:** Abigail Kimball, Lisa Funkhouser-Jones.

**Writing – original draft:** Abigail Kimball, Lisa Funkhouser-Jones, L. David Sibley.

**Writing – review & editing:** Abigail Kimball, Lisa Funkhouser-Jones, L. David Sibley.

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
