## [Decision Letter · Decision Letter 0]

24 Apr 2024

Dear David,

Thank you very much for submitting your exciting Research Article entitled 'Mendelian segregation and high recombination rates facilitate genetic analyses in Cryptosporidium parvum.' to PLOS Genetics.

The manuscript was fully evaluated at the editorial level and by three independent peer reviewers. The reviewers appreciated the attention to an important topic but identified some concerns that we ask you address in a revised manuscript. I apologize for the delay in returning this to you.  Two of the reviewers were a bit delayed beyond the deadline, and it took a couple of additional weeks for the third review to arrive. 

Given the topic of the paper, I have also read it over myself.  One area that I think needs revision has to do with the expectation/prediction for segregation of two unlinked markers following a standard meiosis.  From yeast genetics we know that segregation of two unlinked markers (and which are not centromere linked), will yield parental ditypes (PD), nonparental ditypes (NPD), and tetratypes (TT), if all four meiotic products from a given tetrad are analyzed.  For two unlinked markers, the ratio is PD:NPD:TT of 1:1:4.  Thus, on average you expect about 66% of the progeny to be tetratype.  Thus, it isn't clear why these are suggested to be rare in Figure 2.  In the actual data, the vast majority of your progeny are TT, in accord with this expectation. They are in fact over-represented, so you want to think about why that is the case.  This was also noted by two of the reviewers.

I don't think you need any additional data from what I can see.  Just revisions to the presentation, models, interpretation.

We therefore ask you to modify the manuscript according to the review recommendations. Your revisions should address the specific points made by each reviewer.

Please let us know if you have any questions while making these revisions.  Thank you for submitting this exciting study to PLOS Genetics!   And again, my apologies for the delay in getting this back to you with reviews.

Yours sincerely,

Joe

Joseph Heitman, MD, PhD

Academic Editor

PLOS Genetics

Geraldine Butler

Section Editor

PLOS Genetics

Reviewer's Responses to Questions

**Comments to the Authors:**

Reviewer #1: This is a highly innovative study that will be an excellent addition to the literature on this parasite.

The manuscript is well written, and the results are statistically relevant, and the figs and tables of a high standard.

One minor point; Could the authors clarify the analysis for quantitating oocyst shedding in mouse experiments which is based on a single sporozoite giving rise to an oocyst (line 400 - the average oocyst count is divided by four, since the oocyst contains four sporozoites). The calculation does not seem to take into account that not all merozoites (the progeny of sporozoites) result in male and female gametes. This may explain why they do not see perfect outcrossing from in vivo experiments.

Reviewer #2: The authors investigated meiotic recombination in the apicomplexan parasite Cryptosporidium parvum. My background is in yeast genetics and I was not familiar with the life cycle of C. parvum before reading this manuscript, but the Introduction provided a good crash course in its biology. It is fascinating that the oocyst, which is equivalent to the ascus in yeast (it contains 4 haploid progeny from a meiosis), is the infectious agent in this organism. The standard laboratory culture protocol for growth of C. parvum cultures (incubation in HCT-8 adenocarcinoma cells) does not induce production of oocysts so it has previously been impossible to study the sexual cycle. However, in 2019 (ref. 10) the authors’ laboratory developed a growth method (ALI cultures) that produces oocytes, and this innovation has enabled them to study the meiotic inheritance of genetic markers in the current manuscript.

The manuscript is first detailed study of genetic crosses in Cryptosporidium. The results indicate that C. parvum outcrosses readily, and that it has a high recombination rate. The work is welcome because it is a significant technical achievement, laying the foundation for future genetic crossing and analysis (e.g. QTL analysis) in this important pathogenic species. Overall, the manuscript was written in a clear style that was accessible to this reader.

Major comment

Perhaps I have misunderstood something, but in the cross in Figure 2, I don’t understand why the authors expect the frequency of Tetratype (TT) oocysts to be low. This cross involves markers that are on two different chromosomes (uprt-mCherry on chr. 1, and tk-GFP on chr. 5) so they should segregate independently, as the authors indeed say on P7 L178. So I don’t understand why the authors then go on to say that the TT class is only possible “if crossover occurs during interphase I near the uprt or tk locus” (L184), and they don’t cite any reference to support this statement. They then observe that 87-89% of the oocysts are TT (Fig. 2B,D) and they say on line 200 that this frequency is surprisingly high (also on line 229).

In yeast genetics, a cross involving two markers on different chromosomes would result in the ratio of asci classes being 1 PD : 1 NPD : 4 TT, (i.e. 66% of the asci will be tetratype), unless each of the markers is tightly linked to the centromere of its chromosome. See Hiten Madhani’s online notes about yeast tetrad analysis, https://studylib.net/doc/8744127/classical-genetics--tetrad-analysis-and

The manuscript doesn’t say anywhere that the UPRT and TK genes are centromere-linked, so I presume that they are not. So shouldn’t the question be whether the observed 87-89% frequency is statistically higher than the expected 66% ?

Minor comments

Lines 36-37: As a non-specialist, I did not know what an oocyst is until I read the Introduction, so I didn’t initially understand these lines in the Abstract. It would be helpful if you could modify line 36 to say something like “Infection of ALI cultures or Ifngr1 mice with mCherry and GFP parasites resulted in cross-fertilization and formation of “yellow” oocysts (clusters of 4 haploid cells segregating from a meiosis)”.

Lines 120-122 make analogies to classical genetic analysis in yeast, referring to “random oocyst analysis” and “oocyst tetrad dissection”. It would be nice to return to these analogies in the Discussion to discuss them in more detail. The experiment in Figure 1 is random oocyst analysis which in yeast would be equivalent to random ascus analysis (not random spore analysis). The experiment in Figure 2 is oocyst tetrad dissection, OK. The experiments in Figures 3 and 4 are, I think, equivalent to random spore analysis.

Line 183, “parental non-ditype” should be “non-parental ditype”.

In Table 1, please add two columns showing “% of TT oocysts” and “% of TT + other oocysts”.

P11 L275, spell out Cp as “C. parvum”.

Reviewer #3: Mendelian segregation and high recombination rates facilitate genetic analyses in Cryptosporidium parvum

In this paper, isogenic lines of Cryptosporidium parvum were fluorescently tagged with mCherry or GFP on chromosomes 1 or 5 and allowed to undergo hybridization to investigate the frequency of crossing over, and whether chromosomes segregate independently in a Mendelian manner. To do this, the authors took advantage of a method they developed to culture all stages of Cryptosporidium in stem cell derived intestinal epithelial cells grown under air-liquid interface conditions (ALI), which they published in Cell Host and Microbe in 2019. Growth in ALI cultures generates infectious oocysts that are “yellow”, products of cross-fertilization between a red and green parental line. By FACS purifying “yellow” oocysts, and plating these oocysts in conditions that allowed for individual sporozoites from each oocyst to be counted, a technique the authors dubbed as “oocyst tetrad dissection”, the phenotypes of each individual sporozoite from single oocysts were quantified. The authors also produced additional lines that possessed tags that were linked (on the same chromosome), to calculate crossover frequencies, and to determine a recombination rate for each of the two chromosomes queried, which they estimated at between 3-12.1 kb/cM. This work is significant because the researchers applied new assays to study meiosis in Cryptosporidium and they showed, by examining progeny from crosses between transgenic parasites, that Cryptosporidium utilizes both self-fertilization and cross-fertilization, has high rates of recombination, and that chromosomes segregate in a Mendelian manner. In an effort to compare / contrast their results with those of fungi, in which tetrad analysis is pursued as a genetic technique to study recombination events and meiosis, the researchers used terminology such as parental ditype (PD), non-parental ditype (NPD) and tetratype (TT), and specifically showed that yellow oocysts, generated by fertilization between a gamete that had the red fluorescent tag with a gamete that had the green fluorescent tag, largely produced (ranging from 85-89%) an equal frequency (or 1:1:1:1 tetratypes) of progeny that were either dark, red, green, or yellow. This was expected. However, the researchers found this to be unexpected (see Fig 2A), which appears to be a flaw in their logic and needs to be corrected, or better explained.

Major Points:

1. Figures – are generally well received, and the graphics are excellent summaries of the data, but are lacking the visual display of the raw data. In referring back to the CHM paper in 2019, the figures in that manuscript were admixed with raw data plus summarized results. I am not necessarily asking for the same here, but more extensive experimental data could be added to the supplemental section, as seen for example in Suppl. Fig. 1, which provided real insight into some of the nuances of the dataset, and how timing is critical for data collection, to avoid the complications associated with a second round of meiosis, or the time needed post meiotic reduction (14hpi) to sufficiently dilute the inherited cytosolic fluorescent protein of the other parent, to establish which germline encoded fluorescent protein remained stably expressed for a more precise tabulation in the results.

2. Figure 2A – the logic appears to be flawed, and needs to be better explained, and/or corrected. Cryptosporidium is haploid, no genome reduction occurs to produce gametes, rather a developmental queue commits each haploid parasite to differentiate into either a microgamete or a macrogamete, and upon fusion, a 2n zygote forms. There are no perceived mating types, and from the analyses in this paper, there appears to be an equal frequency of self- vs. cross-fertilization occurring. Hence, there is a 50% chance of being either red or green, or 50% chance of being yellow for the two unlinked markers. And for the oocyst tetrad dissection analysis, the predicted outcome would be a 1:1:1:1 ratio in this scenario, hence dark:red:green:yellow progeny (or TT) from each single, yellow oocyst. This is largely what has occurred (Fig 2B and Fig 2D), and is what is predicted, so it is unclear why Figure 2A depicts the opposite, and a paragraph of text is devoted to this as an unexpected result. Minor deviations from this were evident in the results shown. It was the same whether oocyst formatio

---

## [Editor Report · Decision Letter 1]

27 May 2024

Dear David,

We are pleased to inform you that your revised manuscript entitled "Mendelian segregation and high recombination rates facilitate genetic analyses in Cryptosporidium parvum." has been editorially accepted for publication in PLOS Genetics. Congratulations!  I appreciate the care and attention to detail in the revision process.  I would leave it to you to consider whether you might like to thank the reviewers for their astute insights about tetratype recombinants, and how to frame that part of the interpretation of the findings.  They were right on the mark, and I think this is a very nice example where peer review worked to enhance the paper.

Thank you again for supporting open-access publishing; we are looking forward to publishing your work in PLOS Genetics!  I am especially pleased to see this outstanding study on sexual cycle of Cryptosporidium published in PLOS Genetics---a major advance for the field!

Yours sincerely,

Joe

Joseph Heitman, MD, PhD

Academic Editor

PLOS Genetics

Geraldine Butler

Section Editor

PLOS Genetics

Comments from the reviewers (if applicable):

**Data Deposition**

http://datadryad.org/submit?journalID=pgenetics&manu=PGENETICS-D-24-00143R1

**Press Queries**

---

## [Editor Report · Acceptance letter]

11 Jun 2024

PGENETICS-D-24-00143R1 

Mendelian segregation and high recombination rates facilitate genetic analyses in <i>Cryptosporidium parvum<i>. 

Dear Dr Sibley, 

We are pleased to inform you that your manuscript entitled "Mendelian segregation and high recombination rates facilitate genetic analyses in <i>Cryptosporidium parvum<i>." has been formally accepted for publication in PLOS Genetics! Your manuscript is now with our production department and you will be notified of the publication date in due course.

With kind regards,

Lilla Horvath

PLOS Genetics

On behalf of:
